# A Planning Framework for Robotic Insertion Tasks via Hydroelastic Contact Model

Lin Yang , Mohammad Zaidi Ariffin , Baichuan Lou , Chen Lv and Domenico Campolo *

Robotics Research Center, School of Mechanical and Aerospace Engineering, Nanyang Technological University, Singapore 639798, Singapore; yang0752@e.ntu.edu.sg (L.Y.); mohd.zaidi@ntu.edu.sg (M.Z.A.); baichuan.lou@ntu.edu.sg (B.L.); lyuchen@ntu.edu.sg (C.L.)
* Correspondence: d.campolo@ntu.edu.sg; Tel.: +65-6790-5610

**Abstract:** Robotic contact-rich insertion tasks present a significant challenge for motion planning due to the complex force interaction between robots and objects. Although many learning-based methods have shown success in contact tasks, most methods need sampling or exploring to gather sufficient experimental data. However, it is both time-consuming and expensive to conduct real-world experiments repeatedly. On the other hand, while the virtual world enables low cost and fast computations by simulators, there still exists a huge sim-to-real gap due to the inaccurate point contact model. Although finite element analysis might generate accurate results for contact tasks, it is computationally expensive. As such, this study proposes a motion planning framework with bilevel optimization to leverage relatively accurate force information with fast computation time. This framework consists of Dynamic Movement Primitives (DMPs) used to parameterize motion trajectories, Black-Box Optimization (BBO), a derivative-free approach, integrated to improve contact-rich insertion policy with hydroelastic contact model, and simulated variability to account for visual uncertainty in the real world. The accuracy of the simulated model is then validated by comparing our contact results with a benchmark Peg-in-Hole task. Using these integrated DMPs and BBO with hydroelastic contact model, the motion trajectory generated in planning is capable of guiding the robot towards successful insertion with iterative refinement.

**Keywords:** peg-in-hole assembly; motion planning; contact tasks; Dynamic Movement Primitives; Black-Box Optimization; hydroelastic contact model; bilevel optimization





## 1. Introduction

In recent decades, robot manipulators have been increasingly utilized for various manipulation tasks. In particular, part insertion, e.g., USB port and socket, figures in both daily life and industrial settings [1]. These scenarios highlight the potential of robots in handling contact-rich manipulation tasks. However, the interaction between robots and objects involving contact force is complex, which can lead to manipulation failure, and even damage the device [2]. Hence, it becomes challenging for robot manipulator planning and control [3]. One specific challenge arises from the limitations of visual uncertainty [4], which is widely used in robotic systems. For instance, the vision information cannot always perform well in contact-rich tasks, since some inaccurate pose estimation may cause planning in a mismatch trajectory and finally, failure in an experiment [2]. Therefore, there are difficulties for robots to find a solution for contact-rich tasks, particularly when robotic systems rely solely on vision-based guidance. In order to address this challenge, researchers are exploring the integration of vision sensors with force feedback information [5].

Researchers have proposed many learning-based methods to complete such contact-rich tasks. However, there still exist notable challenges because of the requirement for a substantial amount of sampling and exploration to gather sufficient data [1]. For instance, Levine et al. [6] deployed numerous robot manipulators in the real world, which is time-consuming and expensive. Alternatively, some other researchers prefer training in the

simulator to prepare learning-based algorithms; this methodology works depending on the situation [7]. For example, Blender can only easily simulate kinematics motion of a robot. On the other hand, when dynamics are required, Bullet is widely used for its collision detection, which is a point-contact model. Open Dynamics Engine (ODE) is the default engine of Gazebo, which is famous for its fast calculating speed, but the accuracy is low. Chrono [8] mainly focuses on mobility simulation. MuJoCo [9] is similar to Bullet but has better speed and accuracy. PhysX also focuses on speed instead of accuracy, and is mainly used in video games. Isaac [10] utilizes this engine and is also famous for generating simulated images in a virtual world. Indeed, it is cost-efficient to use the simulators above in non-contact tasks or relatively simple contact tasks. However, as the complexity of contact increases, these simulators may fail or introduce distortions due to inaccurate contact feedback, which means there still exists a significant sim-to-real gap [1]. For instance, an algorithm can be trained with even 100 percent success rate in simulators but still meet failures when deployed into real applications [11]. Alternatively, some researchers are addressing the issue by pursuing highly accurate contact models, such as ANSYS with finite element analysis (FEA) techniques [12]. However, it is impractical to execute training and learning through ANSYS due to the long calculation time associated with FEA. Furthermore, a data-driven method can also be utilized to model contact mechanics. For instance, Peng et al. [13] and Ma et al. [14] utilize neural networks, inputting a substantial amount of real-world data, to accurately regress contact mechanics. However, these approaches may lack the capability to generalize the contact mechanics to any user-defined scenario with varying parameters and geometry shapes. Gathering a comprehensive dataset covering all possible cases can be challenging. Consequently, achieving a balance between accuracy and efficiency remains a challenge when gathering data for contact-rich tasks. It is crucial to find alternative approaches that can accelerate the data collection process without sacrificing accuracy.

Motion planning plays a pivotal role in robotics contact-rich insertion to ensure a safe path to the desired position. Classical motion planning can be divided into these three categories: classical approaches, heuristic approaches, and graph search approaches [15]. Classical approaches, such as potential field, have been traditionally used in motion planning. However, these methods often suffer from limitations in terms of global optimization and robustness. Heuristic exploration can overcome this disadvantage to some extent. Graph Search, such as Astar, is inefficient in complex environments. Moreover, these motion planning algorithms always focus on collision-free motions [16] which are not suitable for complex contact-rich tasks.

Moreover, optimization plays a crucial role in assisting robots to find the best trajectory under specific criteria or constraints, and optimization is widely used in robot planning [2]. For instance, Wang et al. [17] utilize classical optimal control techniques for trajectory planning on a flight deck, which focuses on a collision avoidance problem. However, classical optimization methods may struggle with multi-modal or high-order nonlinear problems [18], because in the real world, contact dynamics is inherently nonlinear and complex, making it challenging to model the contact-rich tasks. The challenge stems from the complexity involved in creating an analytical model that accurately represents the dynamics of contact [19]. Indeed, to address these challenging problems, researchers propose many advanced optimization algorithms. Kurtz et al. [20] take an implicit contact force into account, which is computed at each time step. To specify, they use Iterative Linear Quadratic Regulator (iLQR) to optimize this contact-implicit trajectory. Wei et al. [21] optimizes robot position, speed, and torque, which are used for virtual spring to keep a minimal impedance force and vibration by the forgetting factor function. Moreover, some researchers combine bilevel optimization which combines multiple layers of optimization methods for one optimization problem. For instance, Stouraitis et al. [22] employed a bilevel optimization approach that combines graph search and trajectory optimization to accomplish a collaborative task. Furthermore, robot learning with policy parameterization also utilizes optimization, where policy optimization consists of derivative free/evolutive

(e.g., BBO) and policy gradients (e.g., REINFORCE [23]), and dynamic programming, consisting of policy iteration and value iteration. For example, Suomalainen et al. [3], summarized robot learning methods in contact tasks, including Learning from Demonstration (LfD) and Reinforcement Learning (RL). LfD, such as Gaussian mixture models (GMM) and Gaussian mixture regression (GMR), enable the robot to learn from the demonstrated trajectories provided by human experts. Enayati et al. [24] use GMM to encode human demonstrations and use GMR to generate trajectory, where the policy is parameterized and optimized. However, the original solution of GMM and GMR cannot deal with contact-rich tasks well, which can be improved by RL. For example, an expert's demonstration can be used as a starting point for RL-based policy optimization and minimize the difference between robot and expert [25]. Furthermore, DMPs are widely and popular used in policy parameterization [26], which can be optimized. For instance, Abu-Dakka et al. [4] propose making use of human demonstration to get a peg-in-hole task encoded by DMPs, which helps the robot come to new tasks without new coding.

Peg-in-hole assembly covers many specifics of common contact-rich tasks in various real-world applications. Due to their practical relevance and complexity, numerous researchers have focused their efforts on this. For instance, Whitney et al. [27] consider the whole insertion as quasi-static, then use Newton's law to derive the relation between contact force and pose of peg. Pitchandi et al. [28] also refer to Whitney's work, but add viscoelastic property of compliant material into consideration and derive the appropriate device with optimal stiffness and damping parameters. When the exact pose of hole is uncertain, Lee et al. [29] use a quasi-static model to derive where the hole is based on feedback friction force and geometry of peg. They divide the task into many phases separately. Wu et al. [30] also use equilibrium condition; they focus on robot assembly of circular-rectangular compound peg and hole parts, and use flow chart of hole searching, whose essence is a super if-else machine. Salem et al. [31] insert the hole to the fix peg using quasi-static model by deriving two-point and three-point contact static equations. They also define the assembly sequence by approaching two-point contact, adjusting the direction based on force feedback, until they reach a stable three-point contact phase. However, a geometry method [3] can also work for contact-rich tasks. However, a for geometry method, we need the shape of objects and the environment. Therefore, it is hard to build an analytical model for contact [32].

The contributions of our work are several-fold. Firstly, we enhance the accuracy of simulation by employing the hydroelastic model, which has the potential to reduce the sim-to-real gap. We validate the hydroelastic model against a benchmark to demonstrate its effectiveness. Secondly, we propose a bilevel framework that combines DMPs and BBO, utilizing a hydroelastic model, while this framework incorporates contact force information into the optimization process. Thirdly, we address the inherent uncertainty in visual perception to some extent by introducing noise into the cost function. This allows our framework to adapt to visual uncertainty.

The remaining sections of this paper are structured as follows. Section 2 offers a comprehensive model description and introduces the well-established Whitney's theory. Subsequently, a comparative analysis between the hydroelastic model and Whitney's theory is conducted to assess their performance and identify disparities. In Section 3, our proposed bilevel framework is presented. We explain the methodology that combines DMPs and BBO with the hydroelastic model. This approach aims to tackle the challenges posed by visual uncertainty, which often proves to be a significant obstacle in conventional robot planning solutions. Section 4 concludes the paper and outlines avenues for future research.

## 2. Dynamics Model and Hydroelastic Model

In this section, we first introduce the peg-in-hole scenario. Next, we present a concise mathematical description of our scenario, and introduce the benchmark proposed by Whitney [27]. Subsequently, we extract the curves containing contact force and kinematics information to compare with Whitney's benchmark. By conducting this comparison, we

evaluate the accuracy of the hydroelastic model while also identifying its shortcomings. This evaluation can guide us in effectively utilizing the hydroelastic model.

### 2.1. Dynamic Model

We utilize a hydroelastic model supported in DRAKE [33], a simulator that considers surface contact and offers fast calculation speeds. In our scenario, a virtual robot guides the peg towards a hole, accounting for friction. The term "virtual robot" refers to a robot lacking a physical body but possessing kinematic properties. Meanwhile, a spring connects the ghost robot and peg, as depicted in Figure 1. The initial frame of the peg remains fixed, while users can set the initial frame of the hole, assuming the robot is equipped with sensors, such as a camera, to perceive the pose of the hole. Consequently, the target pose of the peg is known but subject to noise. Therefore, the motion of the peg only depends on the force exerted by the robot and the contact force between the peg and hole. Whenever the contact between peg and hole occurs, the hydroelastic model will generate a surface contact force acting on the peg.

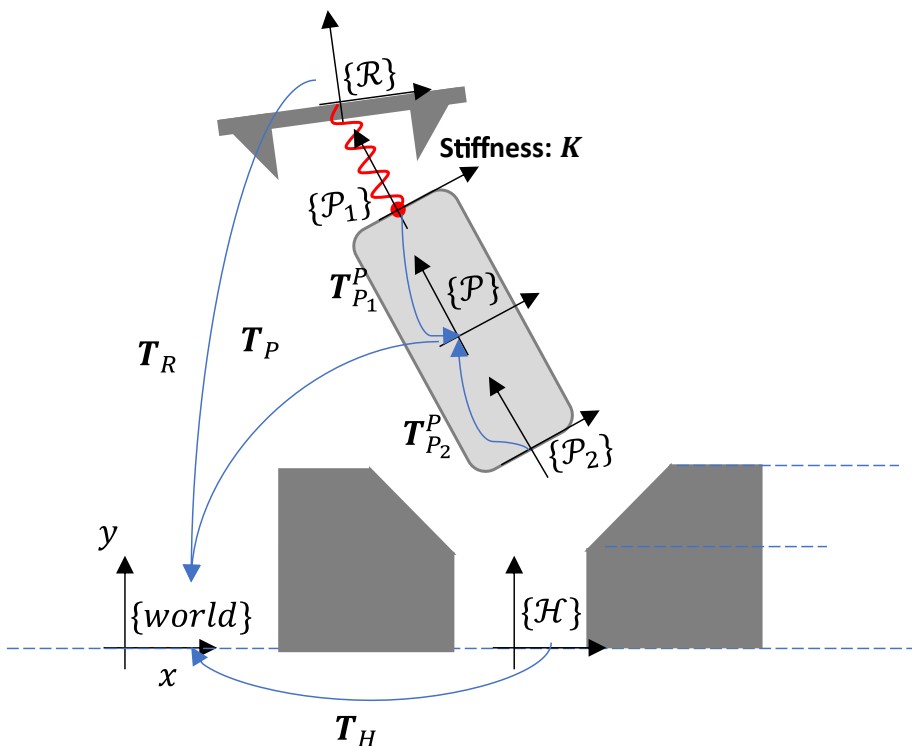

**Figure 1.** Environment description.

We account for the interaction between robot and peg via a simple linear spring model. Therefore, the force applied on the peg is generated by a spring, which only depends on the pose difference between the robot frame $\mathcal{R}$ and peg top center $\mathcal{P}_1$, Numerically, we make use of a simulator, (DRAKE [33]) depicted in Figure 2, to obtain discrete-time evaluation of the dynamics captured by:

$$(q_P, \dot{q}_P, \ddot{q}_P) \quad = \quad \Phi(q_P, q_R, \mathcal{H}) \tag{1}$$

which denotes the position, velocity, and acceleration of the peg which can be generated by this simulator with inputting the pose of the peg at the previous time step, pose of the robot, and models. Ultimately, a successful insertion is achieved when the distance between the bottom surface center of the peg $\mathcal{P}_2$ and the frame of the hole $\mathcal{H}$ is sufficiently close. This criterion determines whether the peg has been successfully inserted into the hole.

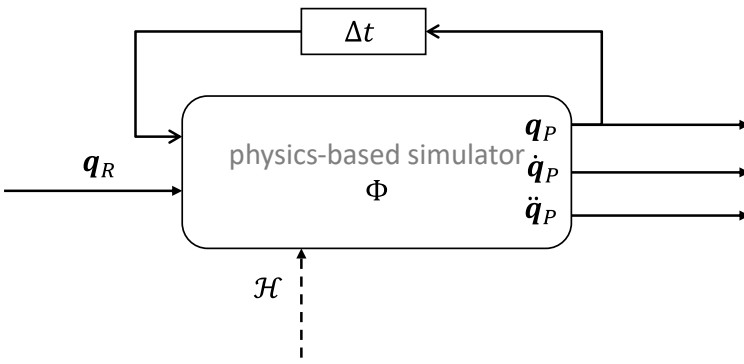

**Figure 2.** Input and output of simulator.

### 2.2. Compliant Interaction

As previously mentioned, it is essential for the robot to apply a force onto the peg during the peg-in-hole task. Here, we shall assume a compliant interaction between robot and peg which, in turn, will allow to evaluate interaction forces from pose differences.

Figure 1 illustrates the scenario we are considering, where $\{\mathcal{P}\} = \boldsymbol{T}_P \in SE(2)$ represents the frame of center of the peg. In addition, we utilize a rotation matrix $\boldsymbol{R}_P$ and transformation matrix $\boldsymbol{T}_P$ to represent the peg frame $\{\mathcal{P}\}$ in the world coordinate system.

$$
\boldsymbol{T}_P = \begin{bmatrix} \boldsymbol{R}(\theta_P) & \begin{matrix} x_P \\ y_P \end{matrix} \\ 0 \quad 0 & 1 \end{bmatrix} \tag{2}
$$

where the general rotation matrix is

$$
\boldsymbol{R}(\theta) := \begin{bmatrix} \cos(\theta) & -\sin(\theta) \\ \sin(\theta) & \cos(\theta) \end{bmatrix} \tag{3}
$$

Similarly, we employ the same definitions for the robot frame $\{\mathcal{R}\}$ and hole frame $\{\mathcal{H}\}$. Furthermore, we introduce two additional frames to define the top surface center of the peg $\{\mathcal{P}_1\}$ and bottom surface center of the peg $\{\mathcal{P}_2\}$. Assuming the peg is a fixed rigid body, we can consider the transformation matrix for $\{\mathcal{P}_1\}$ in $\{\mathcal{P}\}$ to be constant. The length of the peg is symbolized as $l$, $\boldsymbol{T}_{P_1}^P$ equals

$$
\boldsymbol{T}_{P_1}^P = \begin{bmatrix} 1 & 0 & 0 \\ 0 & 1 & \frac{l}{2} \\ 0 & 0 & 1 \end{bmatrix} \tag{4}
$$

The transformation matrix of $\{\mathcal{P}_1\}$ can be expressed as

$$
\boldsymbol{T}_{P_1} = \boldsymbol{T}_P \boldsymbol{T}_{P_1}^P \tag{5}
$$

We employ a function $\boldsymbol{q} = f_{vec}(\boldsymbol{T})$ to extract a vector $\boldsymbol{q}_{P_1}$ to describe $\{\mathcal{P}_1\}$, where

$$
\boldsymbol{q} = \begin{bmatrix} x \\ y \\ \theta \end{bmatrix} = f_{vec}(\boldsymbol{T}) = \begin{bmatrix} T_{13} \\ T_{23} \\ \arctan(T_{21}, T_{11}) \end{bmatrix} \tag{6}
$$

The relative distance between frame $\{\mathcal{R}\}$ and frame $\{\mathcal{P}_1\}$ is written as $\delta \boldsymbol{q}_{RP_1}$.

$$
\delta \boldsymbol{q}_{RP_1} = \begin{bmatrix} x_R - x_{P_1} \\ y_R - y_{P_1} \\ \theta_R - \theta_{P_1} \end{bmatrix} \tag{7}
$$

where $x, y$ denote positions, $\theta$ denotes orientation. Next, we model the interaction between robot and peg by considering three springs with stiffness coefficients $k_x, k_y, k_\theta$ connecting frame $\{\mathcal{R}\}$ and frame $\{\mathcal{P}_1\}$, Additionally, we define the spring between peg and robot in Cartesian space using stiffness matrix $\boldsymbol{K}$:

$$\boldsymbol{K} = \begin{bmatrix} k_x & 0 & 0 \\ 0 & k_y & 0 \\ 0 & 0 & k_\theta \end{bmatrix} \tag{8}$$

Therefore, the elastic interaction between robot and peg can be accounted for by the following energy function

$$E_{RP} = \frac{1}{2}(\delta \boldsymbol{q}_{RP_1})^T \boldsymbol{K}(\delta \boldsymbol{q}_{RP_1}) \tag{9}$$

The elastic (interaction) force exerted by the robot onto the peg $\boldsymbol{f}_{RP}$ can be derived as

$$\boldsymbol{f}_{RP} = -\nabla_{\boldsymbol{q}_P} E_{RP} = -\boldsymbol{K}\delta \boldsymbol{q}_{RP_1} \tag{10}$$

*2.3. Classical Whitney's Model*

Previous derivations are based on a series of simplifying assumptions. In order to verify the realism of such a model, we will benchmark it with classical quasi-static assembly results. Whitney [27] introduced a quasi-static assembly theory for rigid parts utilizing compliant supports, which has undergone comparison with real-world experiments. Their methodology has demonstrated accuracy, making it highly valuable for comparative analysis.

The derivations in the paper involve modelling the compliant supports as springs and deriving equations to describe the forces and displacements during assembly. By analyzing the equilibrium conditions of the system, the derived equations enable predicting the behavior of the parts and supports, offering a flexible approach to assembly processes. Typical insertion geometry has an insertion event with these stages shown in Figure 3: approach, chamfer crossing, one-point contact, and two-point contact.

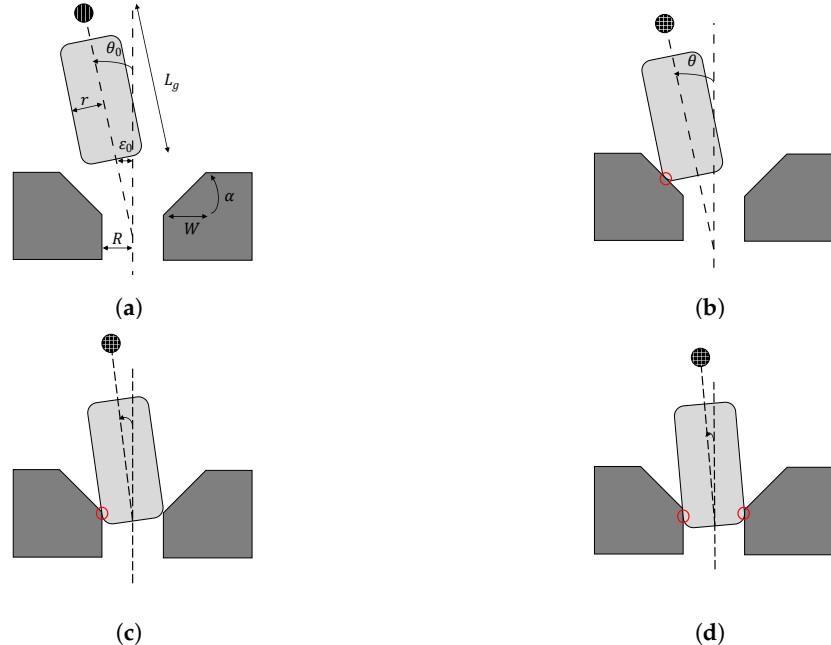

(a)

(b)

(c)

(d)

**Figure 3.** Definition of terms for geometric analysis of part mating. The red circles represent the contact positions. (**a**) Geometry during approach. (**b**) Geometry during chamfer crossing. (**c**) Geometry during One-Point Contact. (**d**) Geometry during Two-Point Contact.

Given the stiffness $K_x$, $K_\theta$, friction coefficient $\mu$ and compliance center $L_g$ with the peg and hole geometrical parameters (such as its initial angular error $\theta_0$, initial lateral error $\epsilon_0$, $c$ clearance ratio), the insertion forces $F_x$, $F_z$ and insertion angles $\theta$ can be calculated from the derived analytical equations in Table 1.

**Table 1.** Insertion forces and angles for various insertion stages reproduced from [27].

| | |
|---|---|
| Chamfer Crossing (For $l = -z$ to 0) | $F_x = -K_x(U_o - U)$ $$F_z = \frac{K_x K_\theta \mathbf{A}(z/\tan\alpha)}{\mathbf{BD} - \mathbf{E}}$$ $$\theta = \theta_0 + \frac{K_x(z/\tan\alpha)(L_g\mathbf{B} - r\mathbf{A})}{(K_x L_g^2 + K_\theta)\mathbf{B} - K_x L_g r\mathbf{A}}$$ |
| One-point Contact (For $l = 0$ to $l_2$) | $F_x = -K_x(U_o - U)$ $$F_z = \frac{\mu K_x K_\theta(\epsilon_0' + \ell\theta_0)}{\mathbf{C}(L_g - \ell) + K_\theta}$$ $$\theta = \frac{\mathbf{C}(\epsilon_0' + L_g\theta_0) + K_\theta\theta_0}{\mathbf{C}(L_g - \ell) + K_\theta}$$ |
| Two-point Contact ($l_2$) (For $l_2$ onwards) | $F_x = -K_x L_g(\theta_0 - cD/\ell) - K_x\epsilon_0''$ $$F_z = \frac{2\mu}{\ell}[\mathbf{D}(\theta_0 - cD/\ell) + \mathbf{F}] + \mu(1 + \frac{\mu d}{\ell}[\mathbf{G}(\theta_0 - cD/\ell) - \mathbf{F}/L_g]$$ $$\theta = \theta_0 + \frac{K_x\epsilon_0''(L_g - l_2 - \mu r)}{K_x L_g^2 + K_\theta - K_x L_g(l_2 + \mu r)}$$ |

where

$$\mathbf{A} = \cos\alpha + \mu\sin\alpha$$

$$\mathbf{B} = \sin\alpha - \mu\cos\alpha$$

$$\mathbf{C} = K_x(L_g - \ell - \mu r)$$

$$\mathbf{D} = K_x L_g^2 + K_\theta$$

$$\mathbf{E} = K_x L_g r\mathbf{A}$$

$$\mathbf{F} = K_x L_g\epsilon_0''$$

$$\mathbf{G} = -K_x L_g$$

$$\epsilon_0' = \epsilon_0 - cR$$

$$\epsilon_0'' = \epsilon_0 + cR$$

*2.4. Comparison between Hydroelastic Model and Whitney's Theory*

To facilitate a comparison, we build models and frame based on Whitney's work [27] within the DRAKE simulator. It is important to note that in this case the peg is moving downward slowly with only a vertical velocity as it approaches the hole.

Furthermore, for defining the material properties of peg and hole in the simulator, we refer the DRAKE's manual [33], which provides guidelines on how to set parameters that correspond to the physical characteristics and behavior of the materials used in the contact-rich tasks. According to the suggestions, we set the hydroelastic modulus same as Young's modulus. Assuming the peg and hole are both made of steel, we set the hydroelastic

modulus as 200 GPA [34]. Then, we set the friction coefficient as 0.6, and Hunt–Crossley dissipation as 30 s/m, which is responsible for the energy-damping property. In order to minimize undesired oscillations of the model during contact, we choose a heuristic value for this parameter.

To facilitate a direct comparison between Whitney's results and the results obtained from the hydroelastic model, we log the contact force and other information by using the same scenario. All simulations were performed on a desktop with an Intel i9 processor and 32 GB RAM.

We are evaluating the quality of the hydroelastic contact model, with the contact force as the performance indicator. Thus, we plot the variation of the contact force with insertion depth. The results are shown in Figure 4 (further results are also reported in Appendix A). We vary the starting position $x_P$ of the peg, which corresponds to changing the horizontal distance or offset between the central line of the hole and the peg. From Figure 4, we observe that the hydroelastic model exhibits the same phases as observed in Whitney's method, albeit with some difference. In the hydroelastic model [33], the horizontal contact force $F_x$ in the peg frame gradually increases during the chamfer contact phase, rather than experiencing a sudden force at the initial contact. As the peg penetrates deeper into the hole, the contact force gradually builds up. After leaving the chamfer phase, there are some oscillations. Subsequently, the peg transitions to one-point contact phase; the value of contact force decreases, then increases, which aligns with Whitney's theory. Finally, during two-point contact phase, we observe that the behavior of the hydroelastic model closely resembles Whitney's theory. The final contact force values converge to very similar values in both the hydroelastic model and Whitney's theory.

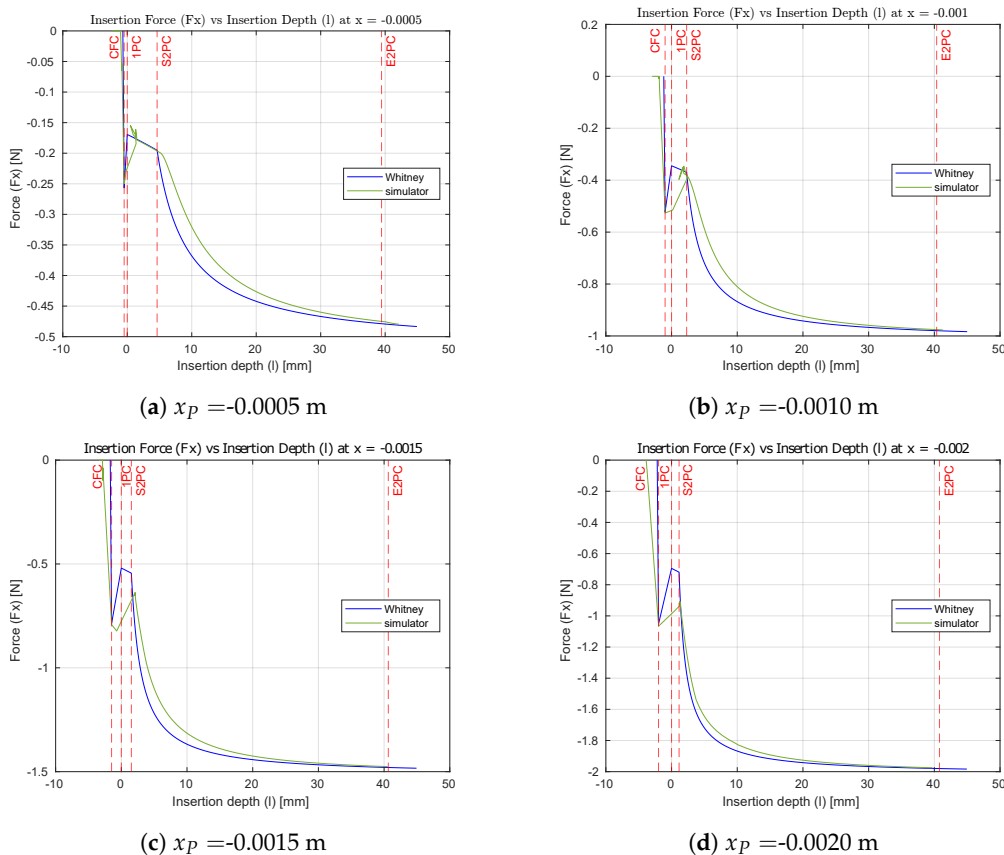

(**a**) $x_P$ =-0.0005 m

(**b**) $x_P$ =-0.0010 m

(**c**) $x_P$ =-0.0015 m

(**d**) $x_P$ =-0.0020 m

**Figure 4.** Insertion Force (Fx) vs. Insertion Depth (l) at different $x_P$ after depth shifting, where CFC symbolizes "Chamfer Contact", 1PC symbolizes "1 Point Contact", S2PC symbolizes "Start 2 Point Contact", E2PC symbolizes "End 2 Point Contact".

Based on the depicted figures, it is apparent that the hydroelastic model consistently reproduces results that closely align with those of Whitney's theory, even when the starting points are altered. The three figures exhibit similar trends and characteristics, thereby supporting the similarity between the hydroelastic model and Whitney's theory; additional results can be found in Appendix A.

We evaluated the relative error at various contact phases for the four situations, and the results are presented in Table 2.

**Table 2.** Relative error between hydroelastic model and Whitney's theory (%).

| Offset | $x_P = -0.0005\,\text{m}$ | $x_P = -0.0010\,\text{m}$ | $x_P = -0.0015\,\text{m}$ | $x_P = -0.0020\,\text{m}$ |
|---|---|---|---|---|
| initial contact force of $F_x$ | 10.7575 | 6.9288 | 0.9780 | 2.9370 |
| final contact force of $F_x$ | 0.2918 | 0.3686 | 0.5088 | 0.8534 |
| initial contact force of $F_z$ | 0.026 | 2.7455 | 5.0961 | 7.8345 |
| final contact force of $F_z$ | 12.7314 | 11.7067 | 10.4138 | 7.3891 |
| final insertion angle | 25.7437 | 22.9141 | 19.2098 | 14.3666 |

Some relative errors are small, within 1%, and the majority of other features are within 10%. However, a few of them exhibit larger relative errors, approximately around 20%. Additionally, it may be emphasized that to get a more comprehensive understanding of how Whitney's results compared with hydroelastic model's results, we need to look at the data throughout the contact task, not just some peak values (as reported in Table 2). Therefore, we introduce this Pearson correlation coefficient to report the overall relationship between the two datasets, denoted as $r$, which is presented in Table 3. The Pearson correlation coefficient measures the strength and the direction of a linear relationship between Whitney's theoretical result and the hydroelastic model's result with possible values between $-1$ and $1$, where a higher value of $r$ indicates a stronger relationship between the two sets of data. Furthermore, the Pearson correlation coefficient can serve as a quantitative measure to assess the extent to which performance in simulations corresponds to performance in the real world [35]. Therefore, we present the linear relationship between Whitney's theory and the hydroelastic model by $F_x$ and $F_z$ at the same insertion depth (after alignment) in Figure 5a,b. In other words, each curve represents the simulated force versus the theoretical force throughout the entire insertion process, with each point plotted at the corresponding shifted insertion depth. We plot a total of four cases, each corresponding to different offsets. Subsequently, we calculate $r$ in Table 3. We can observe that there exists a strong linear relationship between the variables, and some curvatures are caused by the oscillation during contact.

The hydroelastic model demonstrates several advantages. Firstly, it accurately captures the values of the contact force during both the sudden contact and final stable phases, with some relative errors even less than 1%. Secondly, the overall trend of results obtained from the hydroelastic model closely aligns with Whitney's theory, which can be proved by $r(F_x), r(F_z)$; both of them are over 0.9.

However, there are two limitations from the results. Firstly, oscillation occurs when the peg abruptly leaves the chamfer phase. This can be attributed to the virtual spring between robot and peg, which can be considered as stiffness control. When a sudden force disturbs the system, it will result in an oscillation. In this case, when the robot exerts sufficient force to the peg, the peg will suddenly slip off the chamfer and the oscillation phenomenon occurs.

Secondly, there always exists a depth domain delay no matter whether during the chamfer crossing or two-point contact phase. To account for this delay, we shift the results generated by the hydroelastic model based on the depth axis. This is because the contact force arises linearly after instantaneous contact, which is caused by the theory of hydroelastic contact [33]. The insertion depth and contact force will both continue increasing

until the two bodies achieve quasi-static force equilibrium. Therefore, as sudden insertion occurs, we observe a progressive increase in contact force corresponding to the increasing insertion depth, while Whitney's theory makes the contact force suddenly arise. Thus, there exists a depth–domain disparity between these two curves, which explains these curves of the contact results. In plotting Figure 4 and calculating the Pearson correlation coefficient, we aligned the curves based on the peak of first contact force due to the depth domain delay phenomenon. Otherwise, this delay will result in a low Pearson correlation coefficient.

To summarize, the results indicate that the hydroelastic model is capable of accurately simulating the contact-rich behavior, especially with a slow motion that meets quasi-static, and successfully reproducing key characteristics in Whitney's theory, which have already been verified through real-world experiments. Although Whitney's theory is accurate and mature, it fails to account for deformation during contact and lacks the ability to generalize to complex scenarios (e.g., USB port). In other words, when dealing with intricate object shapes, applying Whitney's theory becomes challenging due to the consideration of numerous edges. Consequently, the hydroelastic model not only addresses both deformation and generalization, but also aligns well with Whitney's theory. A user can easily apply the hydroelastic model to any geometry, making it a versatile tool with the potential to support a wide range of application scenarios and provide valuable explanations for real-world contact tasks.

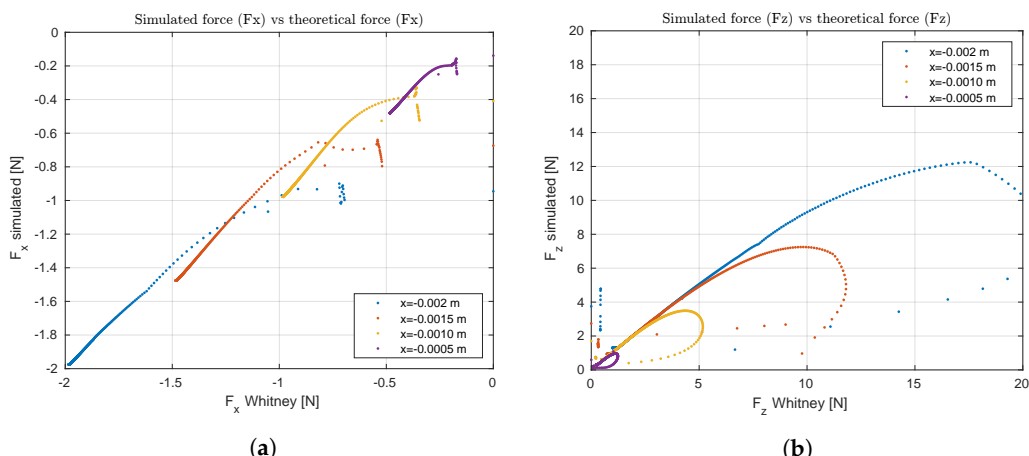

(**a**)          (**b**)

**Figure 5.** Contact force of Whitney's theory vs. hydroelastic model at same shifted insertion depth. (**a**) Horizontal force ($F_x$)—simulated vs. theoretical. (**b**) Vertical force ($F_z$)—simulated vs. theoretical.

**Table 3.** $r(F_x)$ and $r(F_z)$.

| Variable | Pearson Correlation Coefficient $r$ |
|:---:|:---:|
| $F_x$ | 0.9954 |
| $F_z$ | 0.9032 |

## 3. Bilevel Optimization with Hydroelastic Model

With our validated simulation, we can utilize this simulated contact force in the virtual world to optimize our policy for contact-rich tasks, eliminating the need for extensive and time-consuming experiments in the real world. In this section, we will outline the formulation of our planning and optimization method by a peg-in-hole task.

### 3.1. Bilevel Framework

Optimal control is a field of control theory that determines the control input that enables a process to satisfy the physical constraints and minimize some criterion [36]. In the field of robotics, optimal control can be employed to generate optimal trajectories for robots in order to achieve specific goals.

In our problem, the physical constraints arise from the physical contact between the hole and peg. Our objective is to minimize a cost associated with the task. However, calculating the gradient of the contact phenomenon analytically can be complex and challenging. Therefore, in our approach, we utilize BBO, a gradient-free method, to search for the solution that minimizes the cost. BBO provides a flexible and efficient method to explore the solution space without explicitly computing the analytical gradients. By employing BBO, we can effectively optimize the control policy, which is parameterized by DMPs.

The whole computational framework of our approach is illustrated in Figure 6. We propose a bilevel optimization framework. In the inner level, we apply policy parameterization DMPs to plan for trajectory, which is introduced in the Appendix B. One of the salient features of DMP is the definition of low-dimensional parametric vector $\boldsymbol{W}$ to parameterize smooth spatial trajectories. Such a parameter $\boldsymbol{W}$ is then updated during optimization to generate the robot's trajectory. There are several parallel rollouts performed with the varied weight value in the inner level. Additionally, the starting point and target point are provided as inputs to DMPs. The dash arrow represents parameters $\boldsymbol{q}_R(0), \boldsymbol{q}_R(T)$ which will be assumed constant during each rollout of the inner level. After the planned robot trajectory $\boldsymbol{q}_R$ is generated, it is sent to the simulator for execution. The simulator provides the resulting trajectory of the peg, which is then passed to the inner-level optimization with the BBO algorithm. The BBO algorithm analyzes the trajectory and iteratively adjusts the weight values in order to find the best weight. In the outer level, outer optimization sends $\boldsymbol{q}_R(T)$ to DMPs in each iteration. Here, the solid arrow represents $\boldsymbol{q}_R(T)$ which will change in each iteration of outer level. Furthermore, we assume the vision system will detect the environment in each iteration, which represents that $\tilde{\boldsymbol{q}}_H$ is constant in the rollouts but varying during each iteration. $\tilde{\boldsymbol{q}}_H$ is applied in both inner optimization and outer optimization, which affects this task significantly. The subsequent sections will provide detailed explanations of the components depicted in this figure.

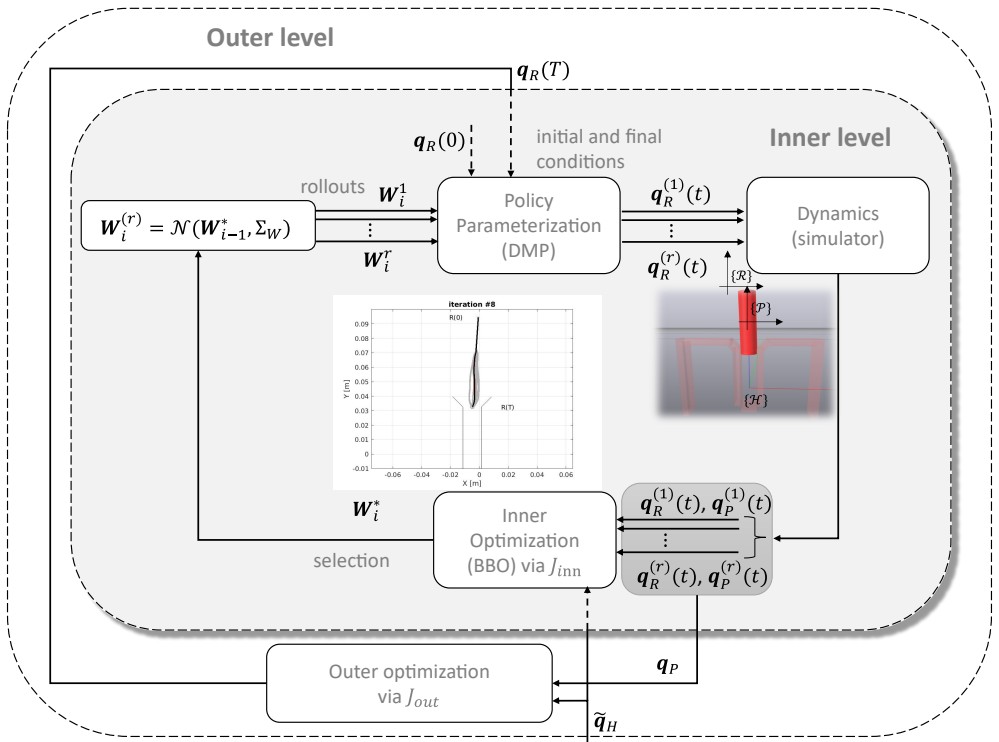

**Figure 6.** Bilevel Framework.

### 3.1.1. Inner-Level Optimization

We choose BBO as the core method for inner-level optimization, because obtaining an analytical description of contact dynamics proves to be challenging. In the inner-level optimization, DMPs require the start and goal points as inputs. As we possess knowledge of the robot's initial pose, we can define the robot trajectory as $t \mapsto \{q_R(t)\}$. By utilizing Equation (9), we can compute $E_{RP}(t)$, while the simulator will record the trajectory of the peg as $q_P(t)$. We employ the same method as Equation (7) to calculate $T^P_{P_2}$, then determine the distance between frame $\{\mathcal{H}\}$ and frame $\{\mathcal{P}_2\}$, which is denoted as $\delta \tilde{q}_{HP_2}$, where visual uncertainty is symbolized as $(\tilde{\cdot})$.

The inner-level cost function is defined as follows:

$$J_{inn}(W, q_R, q_P, \tilde{q}_H) = \alpha_1 \|W\|^2 + \alpha_2 \sum_{t=0}^{t=T} E_{RP}(t) + \alpha_3 \sum_{t=0}^{t=T} (\delta \tilde{q}_{HP_2})^T \Sigma_H^{-1} (\delta \tilde{q}_{HP_2}) \tag{11}$$

where $\alpha_1, \alpha_2, \alpha_3$ are weight coefficients. The first term of Equation (11) represents regularization, which helps in controlling the planning trajectory. The second term represents total energy cost incurred by the robot in the entire insertion process, as calculated in Equation (9). This term encourages the robot to minimize energy consumption during the task. The third term represents the kinematic distance between the current bottom of the peg to the sensed hole, which guides the system move to the target hole. Additionally, $\Sigma_H$ represents the perception error covariance matrix caused by the vision system, which measures the belief associated with the distance between the actual position of the hole and the sensed position of the hole.

In BBO, we perform multiple iterations. During each iteration $i$, we randomly vary the weight parameters $W_i$ according to a Gaussian distribution $\mathcal{N}$ in Equation (A4) with several rollouts, where

$$W_{i+1}^{(r)} = \mathcal{N}(W_i^*, \sigma_W) \tag{12}$$

Subsequently, the trajectory of the robot will be determined by utilizing DMPs Equation (A2), which will then be transmitted to the simulator for execution. Once the simulation is finished, the simulator sends back the simulated results, enabling the calculation of cost for this rollout $J_{inn}^{(r_k)}$ by Equation (11), where a total of $r$ rollouts are considered.

Upon completion of all the rollouts, the rollouts are sorted based on the inner level cost, which is symbolized as $W_{(sorted)(i)}$. Subsequently, the fittest $n\%$ results are selected as the optimal rollout set. The number of elements in the sorted set is $\lceil nr \rceil$, where $\lceil \cdot \rceil$ represents the ceiling function that returns the integer greater than or equal to its input. The weights are then updated based on this selected rollout set.

$$W_{i+1}^* = \frac{\sum_{r_k=1}^{\lceil nr \rceil} W_{(sorted)(i)}^{(r_k)}}{\lceil nr \rceil} \tag{13}$$

Hence, the inner optimization can be written as

$$\min_{W} \quad : J_{inn}(W, q_R, q_P, \tilde{q}_H) \tag{14a}$$

$$\text{s.t.} \quad \ddot{q}_P = \Phi(q_P, q_R, \mathcal{H}) \tag{14b}$$

$$q_R = \text{DMP}(W, t, q_R(0), q_R(T)) \tag{14c}$$

$$t \in [0, T] \tag{14d}$$

where Equations (14b) and (14c) are identical to Equations (1), (A1) and (14d) describe motion duration.

3.1.2. Outer-Level Optimization

In this case, we update the goal $q_{R(i+1)}(T)$ by the exploring result from the previous iteration. The objective is for the goal of DMPs to progressively approach the actual hole frame. Hence, we define the outer-level cost $J_{out}$ as

$$J_{out}(q_{P_2}, \tilde{q}_H) = \frac{1}{2}(\tilde{q}_H - q_{P_2})^T \Sigma_H^{-1}(\tilde{q}_H - q_{P_2}) \tag{15}$$

Therefore, the outer-level optimization is

$$\underset{q_{P_2}}{\arg\min} \quad : J_{out}(q_{P_2}, \tilde{q}_H) \tag{16a}$$

$$\text{s.t.} \quad \tilde{q}_H = [\mathcal{N}(\mu_H, \sigma_x^2), y_H, \theta_H]^T \tag{16b}$$

$$q_{P_2} = f_{vec}(T_{P_2}) \tag{16c}$$

$$T_{P_2} = T_P T_{P_2}^P \tag{16d}$$

$$q_P \in Q_P \tag{16e}$$

where Equation (16b) refers to the sensed hole frame, Equation (16c) represents Equation (6) coming from the simulator, Equation (16d) denotes the two frames in a rigid body, and Equation (16e) describes that $q_P$ is the element of set $Q_P$, which comprises all the peg trajectories in each iteration.

3.1.3. Bilevel Formulation

We combine the two equations presented in Equations (14) and (16) into a bilevel optimization formulation. The pseudocode is shown in Algorithm 1.

---

**Algorithm 1:** Bilevel framework

initialize DMPs, $i = 0$;
**while** $J_{out}$ *not converged* **do**
   $Q_P = \{\}$, Iteration : $i = i + 1$ ;
   Optimize $J_{out}$ : $q_{Ri}(T) \leftarrow \underset{q_{P_1}}{\arg\min} : J_{out}(q_{P_1}, \tilde{q}_H)$ ;
   **for** $r_k \leftarrow 0$ **to** $r$ **do**
      $W_i^{(r_k)} = \mathcal{N}(W_{i-1}^*, \sigma_W)$ ;
      $q_{Ri} = \text{DMP}(W_i^{(r_k)}, t, q_R(0), q_{Ri}(T))$ ;
      $(q_{Pi}, \dot{q}_{Pi}, \ddot{q}_{Pi}) = \Phi(q_{Pi}, q_{Ri}, \mathcal{H})$ ;
      $Q_P.\text{append}(q_{Pi})$
   **end**
   Optimize $J_{inn}$ : $W_{(sorted)(i)} \leftarrow$ sort by $J_{inn}(W_{i+1}^{(r_k)}, q_{Ri}, q_{Pi}, \tilde{q}_H)$ ;
   $W_{i+1}^* = \frac{\sum_{r_k=1}^{\lceil nr \rceil} W_{(sorted)(i)}^{(r_k)}}{\lceil nr \rceil}$
**end**

---

where $q_{Pi}$ represents the whole trajectory of the peg in this iteration.

*3.2. Validation by Peg-in-Hole Task*

We continue to utilize the same scene in the simulator and rely on a camera as being susceptible to inaccuracies. In the peg-in-hole task, the horizontal uncertainty exerts a significant impact. Specifically, when there is a horizontal mismatch between the peg and the hole, it may lead to the peg getting stuck. Therefore, researchers focus on horizontal variation [11]. To account for this, we introduce visual noise for horizontal information $\tilde{x}_H = \mathcal{N}(\mu_x, \sigma_x^2)$ with Gaussian distribution, where $\mu_x = 0.005\,\text{m}$ because the translation

error could reach 5 mm [4], and we also choose $\sigma_x = 0.005$ m to simulate a wide range of uncertainty. The sensed visual information is $\tilde{\boldsymbol{q}}_H = [\tilde{\boldsymbol{x}}_H, y_H, \theta_H]^T$. Regarding the stiffness parameter, it is common to choose 1000 N/m [37]. Similarly, there are no specific requirements for the starting position parameters $x_H, y_H, \theta_H$. Additionally, it is common to consider the coefficients of friction between steel to be around 0.6. The only selected parameters are the weights $\alpha_1, \alpha_2, \alpha_3$. It is important to note that the magnitudes of these three terms in Equation (11) should be similar. If the magnitude of one term is too small, it may not produce the intended effect. Therefore, balancing the weights appropriately is essential to achieve a successful insertion. In practice, we have a rough calibration to compare different weight values for $\alpha_1, \alpha_2, \alpha_3$ and their corresponding terms in Equation (11) in order to select suitable weight values. The initial parameters are detailed in Table 4.

**Table 4.** User-defined parameters.

| Parameter | Value | Units | Description |
|:---------:|:-----:|:-----:|:-----------:|
| $k_x$ | 1000 | N/m | stiffness |
| $k_y$ | 1000 | N/m | stiffness |
| $k_\theta$ | 1 | Nm/rad | stiffness |
| $x_H$ | 0 | m | initial pose |
| $\tilde{\boldsymbol{x}}_H$ | $\mathcal{N}(\mu_H, \sigma_x^2)$ | m | sensed pose |
| $y_H$ | $-0.01$ | m | initial pose |
| $\theta_H$ | 0 | rad | initial pose |
| $\alpha_1$ | 10,000 | - | weight factor |
| $\alpha_2$ | 20 | - | weight factor |
| $\alpha_3$ | 10 | - | weight factor |
| $\mu$ | 0.6 | - | friction |

In the simulation, we performed iterations with 10 rollouts until convergence. Figure 7 displays intermediate results from different iterations. By observing the progression from iteration 1, the initial solution is quite far from the target. However, in subsequent iterations, the policy exhibits a tendency to explore the environment. This improvement can be attributed to the third term in the cost function (11), which evaluates the uncertain distance between the hole and the bottom of the peg. The outer-level optimization in Equation (15) then updates the goal to keep approaching the hole iteratively, motivating to find trajectories towards the uncertain hole against the visual uncertainty.

Around iteration 20 in Figure 7, we observe that the peg has successfully identified the location of the hole but requires further exploration to achieve the bottom. The policy, with the help of the second term of the cost function (11), aims to avoid excessive contact force or getting stuck during the insertion process. As a result, the policy explores different approaches and adjusts its trajectory to achieve a balance between successful insertion and avoiding undesirable contact forces.

In iteration 35 of Figure 7, we observe that the peg has successfully inserted into the hole. At this stage, the policy has refined its trajectory by discarding the exploratory curves from previous iterations. The optimization process, guided by both the first and second terms of the cost function (11), aims to generate a trajectory that is more direct towards the goal. The first term encourages the policy to move towards the target, minimizing deviation on insertion path. Meanwhile, the second term continues to ensure that the contact forces remain within acceptable limits and avoid potential stuck or excessive force during the insertion process.

Figure 8a displays the results of the best peg trajectory for each iteration, while Figure 8b showcases the results of the optimal robot trajectory for each iteration. The gray value of the trajectories represents the iteration number, enabling the observation of policy improvement over time. In the shallow trajectories, the robot moves towards the hole, indicating the searching phase when the policy explores the environment. As the trajectories move into the chamfer region, the policy starts to move downwards, indicating the insertion phase. Through the entire optimization process, we can clearly observe the

gradual improvement of the policy, with trajectories becoming more directed towards successful insertion. Hence, we conclude that the iterative optimization approach allows the policy to explore and refine its trajectory, resulting in a more accurate and effective contact-rich insertion.

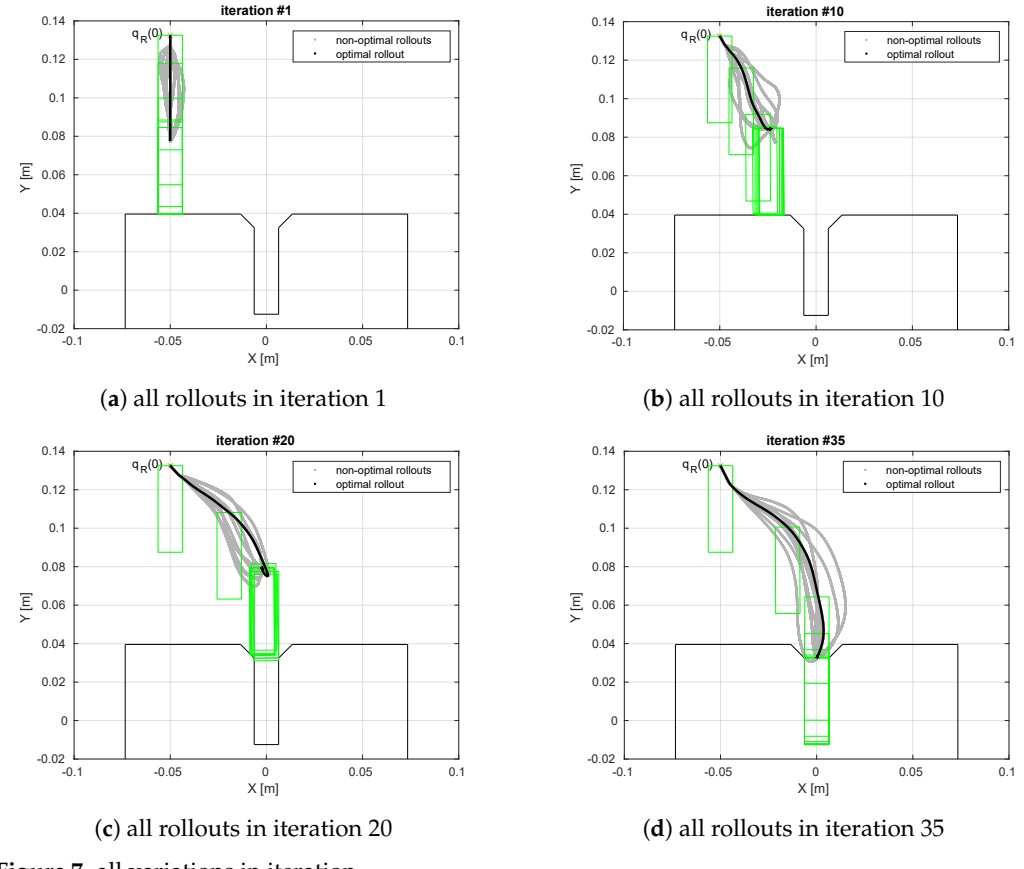

(**a**) all rollouts in iteration 1

(**b**) all rollouts in iteration 10

(**c**) all rollouts in iteration 20

(**d**) all rollouts in iteration 35

**Figure 7.** all variations in iteration.

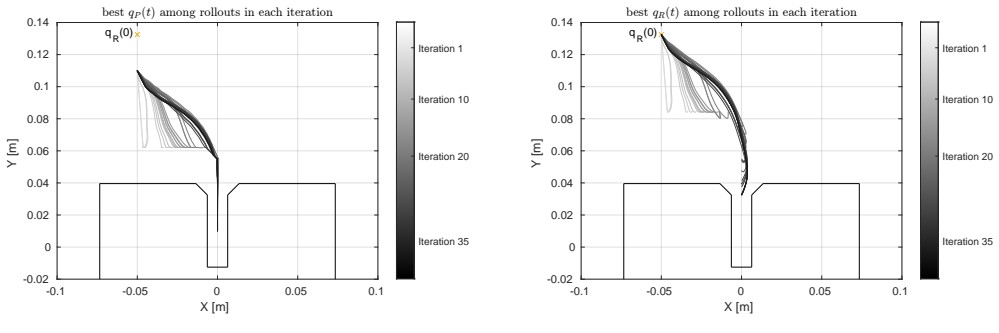

(**a**) iterative optimal peg trajectory.

(**b**) iterative optimal robot trajectory.

**Figure 8.** Optimal trajectory in each iteration.

### 3.3. One-Level Optimization vs. Bilevel Optimization

In order to further validate and evaluate the performance of our approach, we conducted a large-scale simulation with varying starting positions of the peg $x_P = [-0.04 : 0.001 : -0.03]$ for only one-level optimization (only $J_{inn}$) and bilevel optimization ($J_{inn}$ and $J_{out}$). In fact, the routine solution for planning with DMP and BBO is identical to one-level optimization in our framework. For one-level optimization, we directly provide the perceived goal $\tilde{q}_H$ to DMPs, resulting in $q_R(T) = \tilde{q}_H$. To make a fair comparison, we give the same initial goal for our bilevel framework. During the simulation, we run bilevel optimization

until converge, and run same iterations for one-level optimization. The simulation of 50 iterations with 10 rollouts can be finished in 10 min.

The result shown in Figure 9 reveals a consistent trend with our previous findings. We can compare these two methods, the inner-level optimization and one-level optimization. Firstly, in iteration 0, both cases start from the same initial position, and the initial goal given to the DMPs is also the same. Therefore, the cost of the initial problem will be identical for both methods. Secondly, these two cost curves decrease rapidly in two iterations, then the difference between them becomes evident. The one-level optimization method exhibits a converging cost with a high value, indicating that the peg fails to achieve a successful insertion. This reflects the limited effectiveness of the one-level optimization approach in achieving peg-in-hole insertions. On the other hand, bilevel optimization demonstrates convergence of the cost during iterations, reaching a low value, indicating successful insertion tasks. Thirdly, it is worth noting that the effect of visual uncertainty is taken into account in the cost function (11) and (15), which means the final convergence phase may exhibit some mild oscillations due to the inaccuracies representing the uncertain distance between the hole and the bottom of the peg. Therefore, there exist variance bands for both curves. In spite of these variance bands, all the starting points can achieve successful insertions within 50 iterations via bilevel optimization, and the whole trend of cost demonstrates a significant reduction, indicating that bilevel optimization and planning with DMPs can adapt to the uncertainties in visual information, ultimately leading to a successful peg-in-hole insertion.

Thus, we observe that one-level optimization, which is a conventional approach, heavily relies on the accuracy of visual perception and is more susceptible to failure. It struggles to overcome the challenges posed by visual uncertainty, which can hinder the achievement of successful peg-in-hole insertions. In contrast, the bilevel optimization strategy effectively addresses the challenges arising from visual uncertainty. By incorporating both inner and outer-level optimization, it provides a mechanism to iteratively refine the trajectory and adapt to uncertainties in visual information. This enables the cost to converge and ultimately leads to a successful insertion. The bilevel strategy demonstrates its effectiveness by considering different starting positions, which is a reliable approach for achieving successful peg-in-hole insertions.

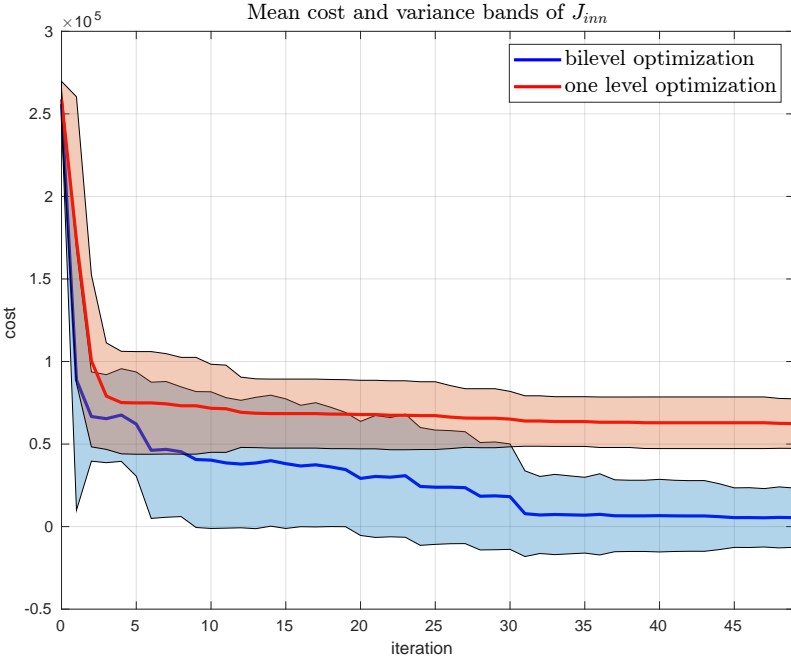

**Figure 9.** Mean and variance bands of one−level optimization and bilevel optimization for different starting positions.

## 4. Conclusions

This paper presents a bilevel optimization framework utilizing DMPs and BBO with a hydroelastic contact model for peg-in-hole tasks. Our research has yielded several key findings. Firstly, we have demonstrated that the hydroelastic model accurately reproduces the essential characteristics of Whitney's theory for a peg-in-hole task. Secondly, we have validated the feasibility of our framework for a peg-in-hole task under visual uncertainty. Thirdly, we have established that conventional planning methods, e.g., one-level optimization, are susceptible to failure in contact-rich tasks due to visual uncertainty. In contrast, our bilevel framework has the capability to overcome visual uncertainty and successfully insert the peg.

Through comparison with Whitney's theory, we have observed that the hydroelastic model generates results with similar features during each contact phase. Notably, the characteristic results achieve $r(F_x) = 0.9954$ and $r(F_z) = 0.9032$. Furthermore, our findings indicate that the hydroelastic model is accurate, with some relative error even less than 1% when motion is slow. The results above prove that the hydroelastic model is a good fit for Whitney's theory, thereby offering potential for reducing the sim-to-real gap. However, when motion is rapid, such as at the sudden contact, the contact force cannot be instantaneously generated. Instead, the surface contact force gradually increases until two complaint bodies reach equal pressure. Therefore, we have also noted the presence of a depth delay phenomenon due to the property of the hydroelastic model.

Our investigations into the bilevel framework with DMPs and BBO have revealed that policy parameterization is a useful and efficient method, where 50 iterations with 10 rollouts can be completed in 10 min. Meanwhile, bilevel optimization guides peg approach to hole with different phases during iteration, which can be logically explained and justified. Specifically, the explored goal for DMPs supported by the outer level has successfully mitigated the visual uncertainty and achieved superior planning trajectories compared with using sensed information directly, where the mean cost of directly using inaccurate hole diverges. Furthermore, our solution demonstrates the ability to achieve 100% successful insertion within 50 iterations across a wide range of starting positions. This showcases the superior performance and reliability of our approach.

For ongoing and future work, we attempt to transfer this planning policy to the real world and test how the hydroelastic model bridges the sim-to-real gap. Additionally, we intend to investigate a more robust framework that can further enhance the effectiveness and reliability of our approach. Moreover, it is worth discussing the generalization of our framework to be applicable to flexible materials. Incorporating another simulator specialized in modeling flexible materials could prove beneficial in extending the versatility of our approach.

**Author Contributions:** Conceptualization, L.Y. and D.C.; Methodology, L.Y. and D.C.; Software, L.Y. and M.Z.A. ; Validation, L.Y.; Formal analysis, L.Y.; Investigation, L.Y.; Resources, D.C.; Data curation, L.Y.; Writing—original draft, L.Y.; Writing—review and editing, L.Y., M.Z.A., B.L., C.L. and D.C.; Visualization, L.Y. and M.Z.A.; Supervision, D.C. and C.L. ; Project administration, D.C.; Funding acquisition, D.C. All authors have read and agreed to the published version of the manuscript.

**Funding:** This research is supported by the National Research Foundation, Singapore, under the NRF Medium Sized Centre scheme (CARTIN). Any opinions, findings, and conclusions or recommendations expressed in this material are those of the authors and do not reflect the views of National Research Foundation, Singapore.

**Data Availability Statement:** The data and results are available in https://github.com/llllllyang/drake-for-Robotic-Insertion (accessed on 14 June 2023).

**Conflicts of Interest:** The authors declare no conflict of interest.

## Appendix A

From Figure A1, which has same trend as Figure 4. The vertical contact force $F_z$ delays at initial contact compared with Whitney, but the peak force values are highly approximate; the relative errors are shown in Table 2, and there is also an oscillation during one-point contact. After coming to the two-point contact phase, the value of $F_z$ increases first then decreases. This behavior is consistent with the expected trend described in Whitney's theory. Then, after coming to two-point contact, the force increases first to a peak, then converges to a highly approximate value.

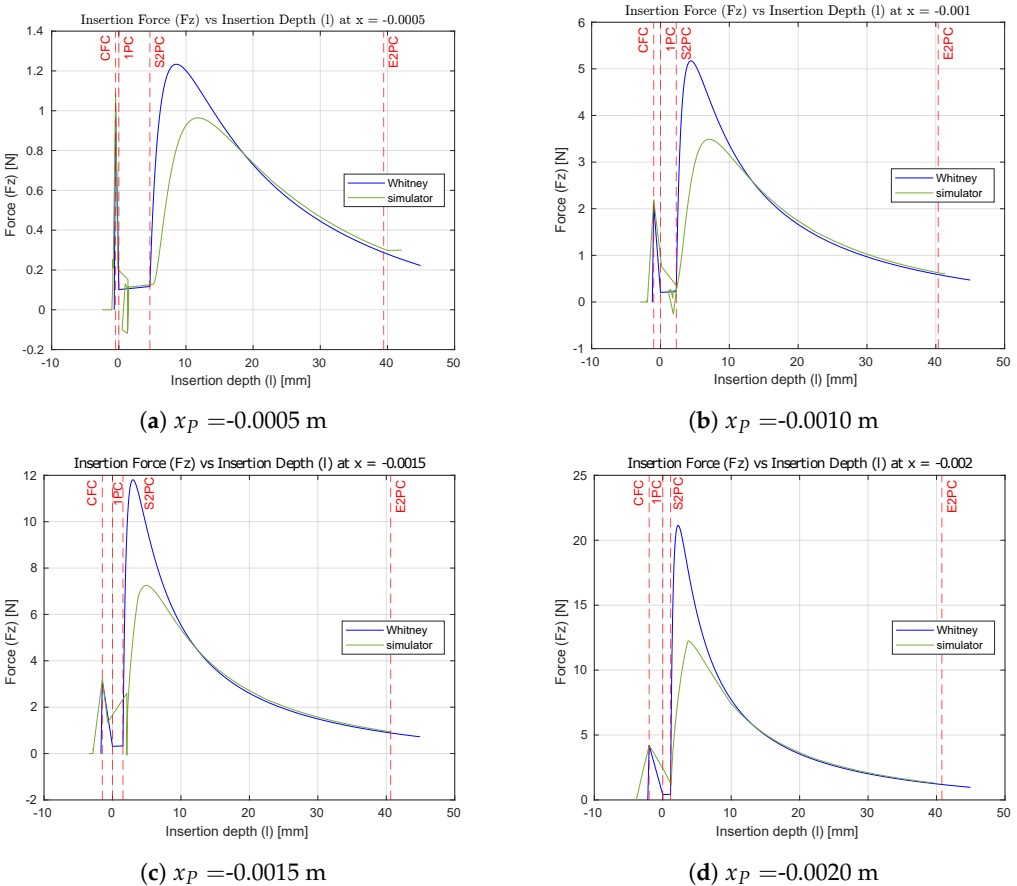

**(a)** $x_P$ =-0.0005 m

**(b)** $x_P$ =-0.0010 m

**(c)** $x_P$ =-0.0015 m

**(d)** $x_P$ =-0.0020 m

**Figure A1.** Insertion Force (Fz) vs. Insertion Depth (l) at different $x_P$ after depth shifting, where CFC symbolizes "Chamfer Contact", 1PC symbolizes "1 Point Contact", S2PC symbolizes "Start 2 Point Contact", E2PC symbolizes "End 2 Point Contact".

From Figure A2. The trend observed in the hydroelastic model is also similar. Although there is a delayed phenomenon compared to Whitney's theory, the final insertion angle can still converge at a value close to Whitney's theory.

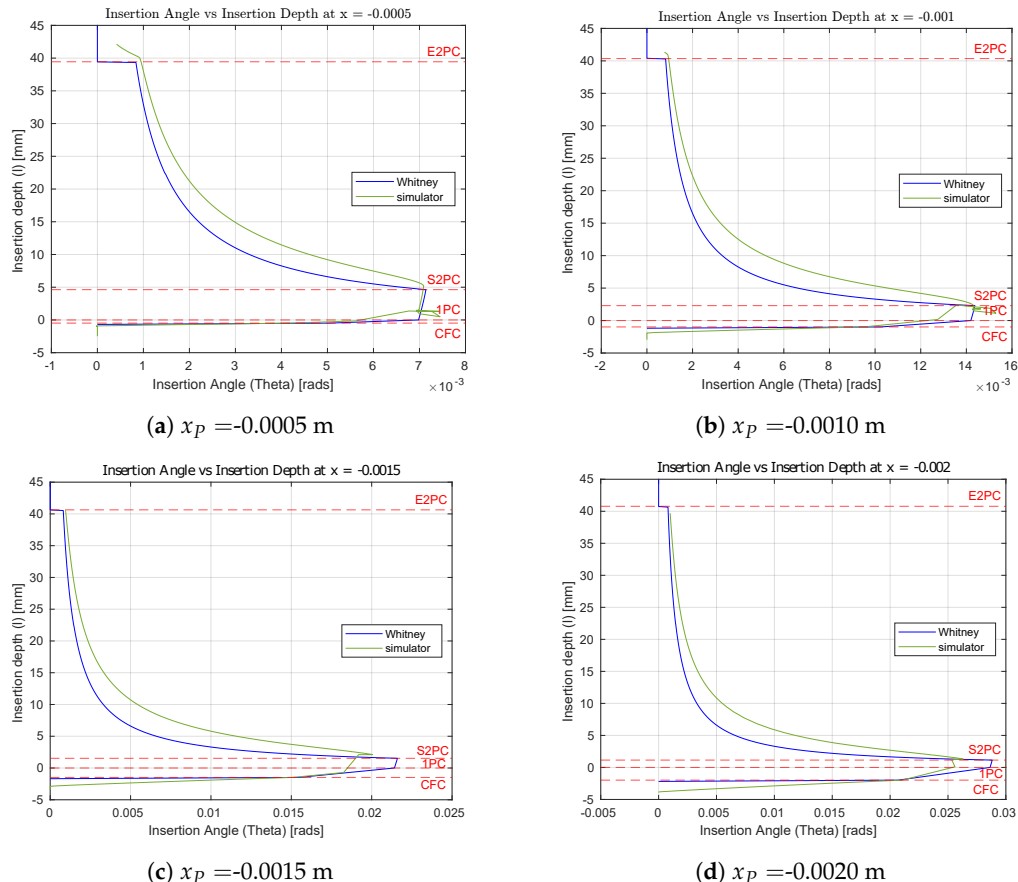

**Figure A2.** Insertion Angle vs. Insertion Depth at different $x_P$ after depth shifting, where CFC symbolizes "Chamfer Contact", 1PC symbolizes "1 Point Contact", S2PC symbolizes "Start 2 Point Contact", E2PC symbolizes "End 2 Point Contact".

## Appendix B

The core idea of DMPs is exerting an external force to a simple linear spring damper system $f_t$, and the force has a nonlinear forcing term $h_W$, which is a function of phase of motion $s_t$. Define the trajectory of robot frame as $t \mapsto \{q_R(t), t = 0 \cdots T\}$. Hence, the start pose of robot is $q_R(0)$ while the goal pose of robot is $q_R(T)$.

Our control input, the robot trajectory, is generated by DMPs,

$$q_R(t) = \mathrm{DMP}(W, t, q_R(0), q_R(T)) \tag{A1}$$

To specify, $q_R(t)$ will be solve as a ODE,

$$\frac{1}{\tau}\ddot{q}_R(t) = f_t + h_W(s_t)s_t(q_R(T) - q_R(0)) \tag{A2}$$

where

$$f_t = \alpha(\beta(q_R(T) - q_R(t)) - \dot{q}_R(t)) \tag{A3}$$

Since in planar motion, each degree of freedom is independent from each other with no constraints, we expand classical DMPs [38] into three dimensions. The output of Equation (A2) is the desired acceleration of the reference trajectory, which means that DMPs can generate a reference trajectory to a nonlinear attractor system with attractor goal $q_R(T)$ and start point $q_R(0)$. Moreover, the second term denotes external force based on the phase of the movement. $s_t$ is the phase and starts at 1 while it ends at 0. This design implies that when external force equals zero, the system will converge to the attractor

$q_R(T)$. Equation (A3) represents a feedback controller (spring-damper system) since it covers current state.

During black-box optimization, $W$ is the parameter that will be optimized based on cost function. The nonlinear forcing term is a function with weight parameter $W$. The general case is multiplying a set of Gaussian kernels. During iteration, the weight will learn from the cost of each trajectory.

$$h(s_t) = \boldsymbol{\phi}(s_t)^T W \tag{A4}$$

$$[\boldsymbol{\phi}(s_t)]_j = \frac{\boldsymbol{w}_j(s_t)}{\sum_{k=1}^{p} \boldsymbol{w}_k(s_t)} \tag{A5}$$

$$\boldsymbol{w}_j = [exp(-\frac{1}{2\sigma_{xj}^2}(s_t - c_{xj})^2), exp(-\frac{1}{2\sigma_{yj}^2}(s_t - c_{yj})^2), exp(-\frac{1}{2\sigma_{\theta j}^2}(s_t - c_{\theta j})^2)] \tag{A6}$$

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
