# Peer review of "A Planning Framework for Robotic Insertion Tasks via Hydroelastic Contact Model"

_machines, doi:10.3390/machines11070741_

Round 1

Reviewer 1 Report

Comments to the Author

 This article presented a motion planning approach with bilevel optimization for robotic insertion. To demonstrate the effectiveness of the proposed approach, simulation results on a peg-in-hole scenario are discussed. The problem is studied comprehensively, but the authors need to address the following comments to improve this work. The reviewer recommends the authors to make major revisions.

 1.In line 64, the author states that "classical optimization methods may struggle with multi-modal or high-order nonlinear problems." But the scale of the simulation examples presented in the manuscript seem not so large, why can't utilize optimal control techniques for solving them? For example, the optimal control method can be used for robot planning and control (https://doi.org/10.1016/j.engappai.2022.105792),

 2. Robot insertion operation must face contact mechanics problems. There are some data driven methods in this field, such as literatures https://doi.org/10.1115/1.4054484. What are the advantages and disadvantages of the contact model adopted by the authors compared with these?

 (https://doi.org/10.1109/TRO.2022.3228390; https://doi.org/10.1016/j.ymssp.2021.107612)

3. There are many parameters in the third section of the manuscript. How are these parameters selected? Are they also applicable to other problems?

 4. When considering the inserted object as a flexible body, can this model mentioned in the manuscript be used? Why?

 5. For peg-in-hole tasks, the comparison between the hydroelastic model and classical Whitney's model identifies some limitations of the hydroelastic model. The classical Whitney's theory has reached a mature stage and undergone comparison with real-world experiments, what specific advantages does the hydroelastic model have over the classical Whitney's model? Why is the hydroelastic model chosen to be employed in this article?

 6. Four images are missing in Figure 3.

none

Reviewer 2 Report

This paper designs a trajectory planning scheme for the problems existing in the current robot insertion tasks that require a large amount of experimental data, etc. And the follow-up simulation results confirmed the effectiveness of the designed framework. But a minor revision is still required, and the final decision can not be made until the following problems are solved.

 The comments are given as follows.

1. Page 4, what is Rθp? And what is the difference between Rθp and Rp ?

2. As a performance indicator, how does the insertion depth reflect the quality of the effect? Is it true that the deeper the depth, the better the effect?

3. In the Billevel Framework, what is the principle of the optimization algorithm of the inner optimizer?

4. In Table 4, why do the values of these parameters vary so greatly, and whether such values are reasonable?

This paper designs a trajectory planning scheme for the problems existing in the current robot insertion tasks that require a large amount of experimental data, etc. And the follow-up simulation results confirmed the effectiveness of the designed framework. But a minor revision is still required, and the final decision can not be made until the following problems are solved.

The comments are given as follows.

1. Page 4, what is Rθp? And what is the difference between Rθp and Rp ?

2. As a performance indicator, how does the insertion depth reflect the quality of the effect? Is it true that the deeper the depth, the better the effect?

3. In the Billevel Framework, what is the principle of the optimization algorithm of the inner optimizer?

4. In Table 4, why do the values of these parameters vary so greatly, and whether such values are reasonable?

Reviewer 3 Report

This paper presents a bilevel optimization framework utilizing DMPs and BBO with hydroelas- 412 tic contact model for peg-in-hole tasks to produce improved trajectories in robot movement to guides peg approach to hole.

Although the paper is well organized, there is minor issues to improve the paper.

1.     The legends in red color of Fig.4 should be clearly arranged. It is hard to read since overlapped by red dotted lines.

2.     In Fig. 7, the legend explaining the differences between black and gray lines should be added.

3.     In Fig. 8, it is hard to tell the trajectory lines corresponding to the legends. Application of different color or line style will be better to address the results in Fig. 8 to readers.

Round 2

Reviewer 1 Report

The revised manuscript can be accepted for publication.

The revised manuscript can be accepted for publication.

Reviewer 3 Report

I think that all the issues identified in the original paper have been successfully addressed.